# AV-CPL: Continuous Pseudo-Labeling for Audio-Visual Speech Recognition

## Abstract

Audio-visual speech contains synchronized audio and visual information that provides cross-modal supervision to learn representations for both automatic speech recognition (ASR) and visual speech recognition (VSR). We introduce continuous pseudo-labeling for audio-visual speech recognition (AV-CPL), a semi-supervised method to train an audio-visual speech recognition (AVSR) model on a combination of labeled and unlabeled videos with continuously regenerated pseudo-labels. Our models are trained for speech recognition from audio-visual inputs and can perform speech recognition using both audio and visual modalities, or only one modality. Our method uses the same audio-visual model for both supervised training and pseudo-label generation, mitigating the need for external speech recognition models to generate pseudo-labels. AV-CPL obtains significant improvements in VSR performance on the LRS3 dataset while maintaining practical ASR and AVSR performance. Finally, using visual-only speech data, our method is able to leverage unlabeled visual speech to improve VSR.

## 1 Introduction

Machine learning has enabled rapid advancement in fields such as speech processing. However, speech processing requires large amounts of labeled data to work well (Radford et al., 2023; Zheng et al., 2022), which is hard to acquire for the thousands of languages spoken world-wide. Semi-supervised learning aims to mitigate this challenge by using unlabeled data to learn better representations and improve performance on labeled data. Real-world unlabeled data is often multi-modal, for example, videos containing synchronized audio and visual information. In this work, we investigate whether we can use such multi-modal data in a semi-supervised pipeline to improve performance on labeled data. Multi-modal data has an additional benefit – modalities can be complementary for each other and provide cross-modal supervision, which influences our algorithm design.

In this work, we study audio-visual speech as multi-modal data with synchronized audio and visual input sequences. Using only the audio or the video data, we can perform two kinds of speech recognition: automatic speech recognition (ASR) from the audio channel, or visual speech recognition (VSR) from the video channel (lip-reading). However, these modalities require substantially different amounts of labeled data for training practical models. For example, with 30 hours of labeled data, we can train an ASR model which reaches around 11% word error rate (WER), while training modern end-to-end VSR models on the same amount of data is challenging: the lowest WER we achieve in our experiments is 96%. Therefore, in this work we investigate how to use the cross-modal information present in audio-visual speech to obtain better VSR performance.

Although VSR is a more challenging task than ASR, VSR still has several useful applications. VSR can be used to transcribe silent videos and to help more people communicate, for example, people with aphonia – a medical condition that causes them to lose the ability to produce voiced sounds (Shillingford et al., 2019). VSR is also useful for audio-visual speech recognition (AVSR) where both the audio and visual modalities are used to predict spoken words. The video channel helps improve performance in noisy audio conditions since it impacted less by background sounds, reverberation, and other distortion (MacLeod & Summerfield, 1987; Afouras et al., 2018a).

In this work, we build upon semi-supervised learning for ASR. So far, there have been two predominant methods: self-supervised learning (SSL) and continuous pseudo-labeling (CPL), or self-training (ST). SSL has two disjoint stages. In the first stage, a proxy task, such as masked recon-

struction, or a contrastive task, is optimized on unlabeled data. In the second stage, the model is fine-tuned on a smaller amount of labeled data (Hsu et al., 2021a; Baevski et al., 2020; 2022; Chiu et al., 2022). CPL instead learns a seed model on labeled data first and then trains the model on labeled and unlabeled data while continuously generating new pseudo-labels on the unlabeled data (Likhomanenko et al., 2021a; Manohar et al., 2021; Higuchi et al., 2021). One of the main benefits of CPL is that it has been shown to match SSL performance with fewer resources, while avoiding the two-stage pipeline by directly optimizing for the downstream task instead of using a proxy task (Likhomanenko et al., 2021a; Berrebbi et al., 2023).

SSL has been applied to audio-visual speech and has been found to decrease the amount of labeled data required to perform VSR and ASR (Shi et al., 2022a; Haliassos et al., 2023; Zhu et al., 2023; Lian et al., 2023). Further, self-training has been applied to audio-visual speech (Ma et al., 2023) and has been found to improve performance when combined with SSL (Shi et al., 2022a; Haliassos et al., 2023). However, current works are restricted to: (i) SSL pre-training and fine-tuning pipeline with two disjoint stages and different objectives; (ii) most of the SSL models are fine-tuned separately for each task (VSR, ASR, and AVSR), which requires $3\times$ the number of model parameters; (iii) self-training is performed *after SSL* pre-training and is often done with an *external ASR model* which itself requires a large amount of labeled data to train, and self-training is done as a single round instead of continuously.

In this work, we propose continuous pseudo-labeling for audio-visual speech recognition (AV-CPL), a semi-supervised method that trains an audio-visual speech recognition model on a combination of labeled and unlabeled data with continuously regenerated pseudo-labels. Our method uses the *same* objective throughout training and can perform ASR, VSR, and AVSR with a *single* model. We use the same audio-visual model for both supervised training and pseudo-label generation, mitigating the need for external ASR models. Our method can handle out-of-domain unlabeled data for self-training with a simple fine-tuning strategy on labeled data. Our approach leads to significant improvements in VSR performance on the LRS3 dataset (Afouras et al., 2018b) while maintaining practical ASR and AVSR performance compared to our baselines trained purely on labeled data. We also conduct a thorough investigation of the training configuration for audio-visual learning, including the architecture design, input stride, and output token set. Finally, we also show that our pseudo-labeling method is effective for unlabeled audio-only and visual-only data.

## 2 RELATED WORK

**Continuous pseudo-labeling for semi-supervised ASR.** Self-training or pseudo-labeling has been successfully applied as a semi-supervised learning method in domains such as vision (Berthelot et al., 2019), machine translation (He et al., 2020), speech recognition (Kahn et al., 2020), and speech translation (Pino et al., 2020). In these methods, a student model is trained on labeled data and is then used to generate pseudo-labels (PL)s for the unlabeled data. For speech recognition, initial methods trained new models from scratch on both the labeled and pseudo-labeled data (Kahn et al., 2020; Xu et al., 2020b; Park et al., 2020; Zhang et al., 2020), sometimes in multiple rounds. They also incorporated a language model (LM) into the PL generation process. However, LM decoding is slower than greedy decoding, and the acoustic models were shown to overfit to the text training set of the LM used for generating PLs. Recent methods such as SlimIPL (Likhomanenko et al., 2021a) and MomentumPL (Higuchi et al., 2021; Manohar et al., 2021) instead continuously train on labeled and unlabeled data while re-generating PLs and use greedy decoding to generate PLs. To prevent model collapse which could happen when PLs are re-generated after each training step, SlimIPL maintains a dynamic cache of unlabeled samples and PLs, while Momentum PL generates PLs with a teacher model whose weights are the exponential moving average of the student model. Inspired by these methods, AV-CPL applies continuous pseudo-labeling for multi-modal speech.

**Semi-supervised learning for AVSR.** The temporal synchrony between acoustic and visual speech provides opportunities for audio-visual semi-supervised learning. Initial methods focused on using external ASR models trained on speech-only datasets for pseudo-labeling unlabeled audio-visual data (Afouras et al., 2020; Ma et al., 2022; 2023) or performing knowledge distillation from the ASR to the VSR model (Li et al., 2019; Afouras et al., 2020; Ren et al., 2021). However, training an ASR model that generalizes well requires a lot of data from different domains (Likhomanenko et al., 2021b; Hsu et al., 2021b), which limits the applications to other languages. AV-CPL does not assume access to any external models and generates PLs continuously by itself while training.

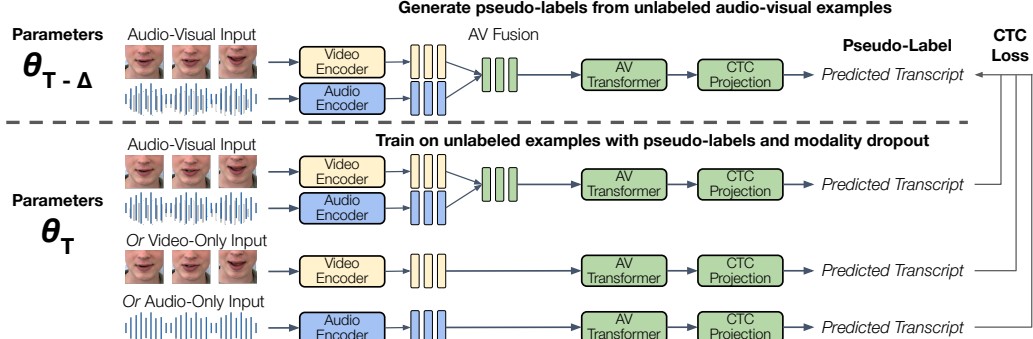

Figure 1: AV-CPL trains jointly on labeled and unlabeled videos while continuously generating pseudo-labels (PL)s on unlabeled videos. The parameters of the model generating PLs, $\theta_{T-\Delta}$, are controlled through a cache or EMA (explained in Section 3.2). Audio-visual inputs are used during PL generation. Modality dropout is used during training so that the model is trained on audio-visual, video-only, or audio-only inputs to increase robustness for missing modalities.

**Self-supervised learning for AVSR.** Recently, SSL has been applied to audio-visual speech to improve VSR using unlabeled audio-visual data. Learning objectives for the proxy task include masked token prediction (Shi et al., 2022a; Hsu & Shi, 2022; Zhu et al., 2023) and predicting the latent representations of teacher models (Ma et al., 2021a; Haliassos et al., 2023; Zhang et al., 2023; Lian et al., 2023). Although most of the models use a shared audio-visual transformer to process both audio and visual inputs, they usually fine-tune separately for each task (VSR, ASR, AVSR), which requires $3\times$ the number of parameters. u-HuBERT (Shi et al., 2022a) is one exception which fine-tunes on audio-visual data and performs all three tasks. Some methods combine SSL with self-training and gain a boost in performance by using an ASR model to label the unlabeled data (Shi et al., 2022a; Haliassos et al., 2023; Zhu et al., 2023). However, the ASR model is *external* and trained separately from the audio-visual model, and pseudo-labeling is done only once. AV-CPL forgoes the proxy task and instead is trained for speech recognition from audio-visual inputs throughout training. It performs pseudo-labeling on unlabeled data continuously during training and only requires a single model to perform all three tasks.

## 3 METHOD

### 3.1 SUPERVISED AUDIO-VISUAL TRAINING

At the core of our method is an audio-visual transformer (Vaswani et al., 2017; Afouras et al., 2018a). Given an input of synchronized audio and visual sequences $\mathbf{A}_{1:T}$ and $\mathbf{V}_{1:T}$, both modalities are first processed by separate encoders, resulting in audio features $\mathbf{f}_{1:T}^a$ and visual features $\mathbf{f}_{1:T}^v$. These features are then added to form audio-visual features: $\mathbf{f}_{1:T}^{av} = \mathbf{f}_{1:T}^a + \mathbf{f}_{1:T}^v$. The audio-visual features are combined with a positional embedding and fed as input into the transformer, which predicts the tokens corresponding to the spoken phrase in the input. We train our models with the Connectionist Temporal Classification (CTC) objective (Graves et al., 2006). Recent audio-visual models adopt the sequence-to-sequence (S2S) framework with a transformer decoder (Ma et al., 2021b; Shi et al., 2022a). The main reason we focus on CTC models instead of S2S models is to avoid issues during pseudo-label (PL) generation due to looping and over/under-generation: it is well known that S2S models tend to generate shorter or longer sequences and are likely to generate repeated $n$-grams at the end of sequences. Consequently, S2S models necessitate strategies for filtering poor PLs (Park et al., 2020; Kahn et al., 2020; Gheini et al., 2023), while PLs generated by CTC models do not require filtering (Likhomanenko et al., 2021a; Higuchi et al., 2021).

By default, the model is trained for audio-visual speech recognition, which uses both audio and visual inputs. However, to increase the model's robustness to a missing modality and to facilitate VSR-only and ASR-only capabilities, we randomly apply modality dropout during training where one modality is entirely dropped out (Neverova et al., 2015; Shi et al., 2022a). Both modalities are used as input with probability $p_m$. If only one modality is used, the audio features are used with

probability $p_a$. Formally,

$$\mathbf{f}_{1:T}^{av} = \begin{cases} \mathbf{f}_{1:T}^a + \mathbf{f}_{1:T}^v & \text{with probability } p_m \\ \mathbf{f}_{1:T}^a & \text{with probability } (1 - p_m)p_a \\ \mathbf{f}_{1:T}^v & \text{with probability } (1 - p_m)(1 - p_a). \end{cases} \quad (1)$$

By default, $p_m = p_a = 0.5$. We always set $p_m = p_a$ for simplicity so that a lower probability means more video-only training, and a higher probability means more audio-only training.

## 3.2 CONTINUOUS PSEUDO-LABELING

Once an initial seed model is trained on the labeled data, we use the seed model for semi-supervised continuous pseudo-labeling (CPL). With labeled $L = \{(\boldsymbol{a}_i, \boldsymbol{v}_i), \boldsymbol{y}_i\}$ and unlabeled $U = \{(\boldsymbol{a}_j, \boldsymbol{v}_j)\}$ videos, the audio-visual model $\mathcal{M}_{\boldsymbol{\theta}}$ is trained on both labeled and unlabeled data with continuously re-generated PLs. The following loss function is minimized during training: $\mathcal{L}(\boldsymbol{\theta}) = \mathcal{L}_L(\boldsymbol{\theta}) + \lambda\mathcal{L}_U(\boldsymbol{\theta})$, where $\boldsymbol{\theta}$ are the model's parameters, $\mathcal{L}_L(\boldsymbol{\theta})$ is the CTC loss using the labeled data, $\mathcal{L}_U(\boldsymbol{\theta})$ is the CTC loss using the unlabeled data and PLs, and $\lambda$ is a hyperparameter controlling the weight of the unlabeled data. To decouple the seed model training from the pseudo-labeling stage, the optimizer is restarted and new modality dropout probabilities $p'_m = p'_a$ are used. To generate PLs for the unlabeled audio-visual data, samples are passed through the model without augmentation and the model's predicted transcripts using greedy decoding are used as the PLs. The model can generate PLs using both the audio and visual data as input (AVSR), or just the audio (ASR) or visual (VSR) modalities. In practice, we use both modalities (AVSR) to generate PLs since the performance is slightly better than ASR, and we wanted to prevent the model from over-relying on the audio input.

We propose two methods to control the generation of PLs: **AV-SlimIPL** (see Appendix, Algorithm 1) and **AV-EMA-PL** (see Appendix, Algorithm 2). AV-SlimIPL maintains a dynamic cache of unlabeled samples and PLs. Before CPL begins, the model is "warmed up" on the labeled data with the new modality dropout probabilities $p'_m = p'_a$. Then CPL begins and the cache of size $C$ is filled with unlabeled audio-visual samples and their PLs generated by the audio-visual model with states from different training iterations. During CPL, the model is continuously trained on samples from the labeled dataset and from the cache of unlabeled samples and PLs. The cache is updated with a probability $p$ (controls the number of PL re-generations) by replacing some samples in the cache with other unlabeled data and their PLs generated by the current model state. This ensures that PLs are updated using newer versions of the model which are improved upon older states.

AV-EMA-PL instead generates PLs with a separate *teacher* model $\mathcal{M}_{\boldsymbol{\phi}}$ whose parameters are updated as the exponential moving average (EMA) of the *student model* $\mathcal{M}_{\boldsymbol{\theta}}$'s parameters. Before CPL, both $\boldsymbol{\theta}$ and $\boldsymbol{\phi}$ are initialized with the parameters of the seed model. The student model is trained with regular gradient-based optimization while the parameters of the teacher model are updated as $\boldsymbol{\phi} \leftarrow \alpha\boldsymbol{\phi} + (1 - \alpha)\boldsymbol{\theta}$, where $\alpha$ is a hyperparameter controlling the weight of the most recent student parameters. During an initial "warm up" phase, the student model is trained on the labeled data with the new modality dropout probabilities $p'_m = p'_a$ and the teacher model's parameters are updated as the EMA of the student's parameters. Once CPL begins, the teacher model generates PLs on the unlabeled data at each iteration and continues to track the EMA of the student's parameters.

AV-SlimIPL and AV-EMA-PL have different computational requirements since the former requires a cache and the latter requires two models. Given that video frames take longer to load and require more memory than audio samples, it is easier to maintain two models instead of a cache (see Appendix A for more details). Therefore, in practice we use AV-EMA-PL for our AV-CPL experiments. However, AV-SlimIPL and AV-EMA-PL are closely related: they both perform model averaging by using previous model states to generate PLs. The $\alpha$ hyperparameter in AV-EMA-PL can be related to the cache size $C$ and probability $p$ in AV-SlimIPL as $C = \frac{p}{1-\alpha}$ to provide a similar model history horizon for model averaging (either via EMA or via the cache). We use $\alpha = 0.9999$ in our experiments which corresponds to $C = 10,000$ and $p = 1$ or $C = 1,000$ and $p = 0.1$; the later is faster to train as PLs are regenerated only every 10th training step.

**Differences with audio-only CPL.** AV-SlimIPL and AV-EMA-PL are inspired by audio-only SlimIPL (Likhomanenko et al., 2021a) and EMA-PL (Higuchi et al., 2021; Manohar et al., 2021) and use similar mechanisms for controlling the generation of PLs. We stress that these previous methods only trained and evaluated using audio, while our models are trained with both audio and visual

inputs and perform ASR, VSR, and AVSR. Our method uses cross-modal information to generate PLs using both audio and visual inputs. Moreover, VSR is more challenging task than ASR: our seed models' VSR performance ($>$60% WER) is higher than the audio-only seed models' ASR performance ($\approx$20% WER). Nonetheless, we are able to improve the VSR performance significantly.

## 4 EXPERIMENTAL SETUP

Following prior works on AVSR, LRS3 (Afouras et al., 2018b) is used as the labeled dataset while VoxCeleb2 (Chung et al., 2018) is used as the unlabeled dataset. LRS3 is the largest public dataset for audio-visual speech recognition in English collected from TED talks. We followed Shi et al. (2022a) to generate the following splits: 433h training set, 30h training set, 1h validation set, and 1h test set. When training the models on the 30h set, the 433h data can also be used as unlabeled data. VoxCeleb2 is a multilingual audio-visual speaker verification dataset without transcripts. While the original dataset contains more than 2,400 hours of video, we use the 1,326 hours of videos in English selected by Shi et al. (2022a). We note that the VoxCeleb2 videos are from a different distribution than those in LRS3 since they were collected from YouTube, are longer, and have more noise. The dataset statistics are reported in the Appendix, Table B1.

Following prior audio-visual semi-supervised learning setups, we use two transformer model sizes: Base and Large. The number of transformer blocks / embedding dimensions / feed-forward dimensions / attention heads for Base / Large are 12/768/3072/12 and 24/1024/4096/16 respectively. We use the CAPE positional embedding (Likhomanenko et al., 2021c). The number of parameters is 96M for Base and 315M for Large. Full training details are presented in Appendix C.

We use the videos' original frame rate of 25 fps (corresponds to 40ms stride per frame). Following Shi et al. (2022a), Dlib (King, 2009) is used to detect facial key points to extract a 96x96 region centered on the mouth. During training, we take a random 88x88 crop and flip the entire video horizontally with probability 0.5. During testing, we use the center 88x88 crop and flipping is not applied. The videos are converted to a single channel (grayscale). The videos are processed with a 3D convolutional layer (Stafylakis & Tzimiropoulos, 2017), followed by a ResNet-18 (He et al., 2016). The audio sampled at 16kHz is converted to an 80-dimensional Mel spectrogram with a stride of 10 ms and a window size of 25 ms. The model processes the spectrograms with a 1D convolution with a stride of 2 (stride is 20ms per output frame). We duplicate the video features temporally so that both modalities have a stride of 20ms. We provide an analysis of the stride in Table 2b.

We train the models with the CTC loss using character output units. We use English characters and numbers, augmented with word boundary, apostrophe and CTC blank token. We train 4-gram word-level language models (LM)s on the LRS3 30h/433h training text using KenLM (Heafield, 2011) and use it with the Flashlight beam-search decoder (Kahn et al., 2022) implemented in Torchaudio (Yang et al., 2022). Full decoding details are presented in Appendix D. We select the best model checkpoints on the validation set to evaluate on the test set and report results on both the validation and test sets. We include a discussion about the performance on the validation set in Appendix E.

## 5 RESULTS

### 5.1 AUDIO-ONLY CONTINUOUS PSEUDO-LABELING

We first conducted experiments on audio-only and video-only continuous pseudo-labeling (CPL) to confirm the effectiveness of the method on each modality before combining them. We show the audio-only CPL results in Appendix F. We re-implemented SlimIPL (Likhomanenko et al., 2021a) as the audio-only CPL method and compared it to HuBERT (Hsu et al., 2021a) as the audio-only self-supervised method, using LRS3 and VoxCeleb2 for labeled and unlabeled data. We found that SlimIPL can outperform HuBERT with a simpler pipeline (CTC encoder model and a 4-gram LM, compared to a S2S encoder-decoder model). The results show that audio-only CPL methods can transfer well to new datasets, motivating us to perform video-only and audio-visual CPL.

Table 1: Comparison of video-only models. AV-HuBERT is a S2S encoder-decoder transformer trained from scratch with video-only, while V-CPL is a CTC encoder transformer. We report either greedy ("None") or beam-search decoding with an LM trained on LRS3 30h or 433h transcriptions.

| Method | Transformer Model | Encoder Size | Criterion | LRS3 Labeled | VoxCeleb2 Unlabeled | Test WER (%) | | |
|---|---|---|---|---|---|---|---|---|
| | | | | | | None | LM 30h | LM 433h |
| *Supervised* | | | | | | | | |
| AV-HuBERT (Shi et al., 2022a) | Large | 325M | S2S | 30h | - | **92.3** | - | - |
| V-CPL (Ours) | Large | 314M | CTC | 30h | - | 103.7 | 96.5 | **96.6** |
| AV-HuBERT (Shi et al., 2022a) | Large | 325M | S2S | 433h | - | **62.3** | - | - |
| V-CPL (Ours) | Large | 314M | CTC | 433h | - | 66.0 | 61.1 | **60.6** |
| *Semi-Supervised with **External ASR** Models* | | | | | | | | |
| AV-HuBERT (Shi et al., 2022a) | Large | 325M | S2S | 433h | 1,326h | **51.7** | - | - |
| *Semi-Supervised with Continuous Pseudo-Labeling (**Ours**)* | | | | | | | | |
| V-CPL (Ours) | Large | 314M | CTC | 433h | 1,326h | **61.0** | 55.9 | **55.9** |

## 5.2 VIDEO-ONLY CONTINUOUS PSEUDO-LABELING

In Table 1, we show the results of applying CPL to the video modality only (**V-CPL**). We use the AV-EMA-PL method without any audio input. We show the full results including Transformer-Base and results on the validation set in the Appendix, Table F2. Training the video-only transformer model on labeled LRS3 30h from scratch is challenging – the best WERs we are able to get is around 96%. When training on labeled LRS3 433h video-only data, we obtain a more reasonable WER of 60.6%. Our results are similar to video-only AV-HuBERT trained from scratch without self-supervised pre-training, although our model is simpler and does not use a transformer decoder. When we apply CPL using 433h labeled video-only data, the 1,326h unlabeled video-only data from VoxCeleb2 improves the WER to 55.9%. These results show that it is possible to perform CPL with unlabeled silent videos even when the seed model has relatively large WER ($> 60\%$). We provide an ablation study for the ratio of unsupervised to supervised updates in the Appendix, Table G1. Our method achieves similar performance to video-only AV-HuBERT trained from scratch using PLs on VoxCeleb2 generated by an external ASR model (51.7%), while our V-CPL method does not use any audio input at all. These results confirm that it is possible to perform video-based pseudo-labeling *without an external ASR model*, motivating our audio-visual pseudo-labeling approach.

## 5.3 AUDIO-VISUAL MODEL DESIGN

In this section, we investigate the best architecture design and training pipeline for supervised AVSR to obtain the best seed model for audio-visual continuous pseudo-labeling. Note that the modality dropout while training the seed model is $p_m = p_a = 0.5$.

**AV Architecture.** In Table 2a, we compare audio-visual architectures. For the audio encoder, Shi et al. (2022a) proposes to stack 4 spectrogram frames with an effective stride of 40ms to match the video frame rate. This is equivalent to a convolutional layer with stride of 4 and kernel width of 4. We tried this method (Linear) as well as a convolutional layer with stride of 4 and kernel width of 7 (Conv.). For the modality fusion method, the audio and visual features can be fused either by temporally concatenating the features and passing them through a linear layer (Shi et al., 2022a), or by adding the features together. We also control whether modality drop is enabled or not. We find that the convolutional layer works better than the linear layer according to the ASR and AVSR performance. For modality fusion, addition works better with the convolutional layer, while the results for the linear layer are mixed. Modality dropout tends to make AVSR and VSR marginally worse, and ASR significantly better. Given these results, we use the convolutional layer with modality addition for all subsequent experiments.

**Token Set.** Two common token sets in AVSR are characters or subwords. Subwords are longer than characters and typically work better with a larger input stride. We first compared different tokens and strides for the audio-only and video-only supervised models in Table G2 and Table G3 of the Appendix. We follow Shi et al. (2022a) to construct unigram-based subwords with a vocabulary size of 1k (Kudo, 2018). We found that the audio model works best with characters and a stride of 20ms, while the video model works better with characters when performing CPL. In Table 2b, we compare different tokens and strides for the AVSR supervised model, where we observe trends consistent with the audio-only and video-only results. When using the AVSR model for ASR, the best results are obtained using characters and a stride of 20ms. For VSR, subwords outperform characters, and using a stride of 20ms works better than a stride of 40ms with characters. Even though the 20ms stride contains the same visual features as the stride of 40ms (duplicated), the

Table 2: AVSR ablation studies with 433h LRS3 labeled videos, Transformer-Base, and 433h LM.

(a) Comparing audio encoders (convolutional vs linear, 40ms stride) and fusion (concat. vs addition).

| Audio Model | Fusion | Mod. Drop | ASR WER Val | Test | AVSR WER Val | Test | VSR WER Val | Test |
|---|---|---|---|---|---|---|---|---|
| Conv. | Concat. | N | 18.7 | 20.0 | 8.5 | 10.1 | 60.4 | 69.6 |
| Conv. | Concat. | Y | 8.0 | 5.9 | 12.7 | 13.5 | 71.3 | 77.4 |
| Conv. | Add | N | 17.9 | 19.6 | **8.2** | **9.1** | 60.2 | **69.0** |
| Conv. | Add | Y | **7.8** | **5.7** | 12.2 | 12.3 | 71.3 | 78.0 |
| Linear | Concat. | N | 32.8 | 31.7 | 9.1 | 10.1 | 62.7 | 70.8 |
| Linear | Concat. | Y | 8.0 | **5.6** | 13.8 | 14.8 | 71.1 | 77.8 |
| Linear | Add | N | 27.7 | 31.3 | 8.6 | 9.4 | 61.1 | 69.4 |
| Linear | Add | Y | 8.6 | 6.0 | 13.9 | 14.4 | 73.0 | 78.1 |

(b) Token set (characters and subwords) vs stride.

| Tokens | Stride | ASR WER Val | Test | AVSR WER Val | Test | VSR WER Val | Test |
|---|---|---|---|---|---|---|---|
| Characters | 20ms | **4.2** | **3.0** | 7.8 | 9.6 | **66.4** | **74.6** |
| Characters | 40ms | 7.8 | 5.7 | 12.2 | 12.3 | 71.3 | 78.0 |
| Subwords | 20ms | 6.3 | 9.9 | **5.6** | **9.7** | 67.6 | 72.4 |
| Subwords | 40ms | **5.5** | **7.1** | 8.4 | 14.4 | **60.6** | **70.1** |

(c) Pre-training on audio/video-only (20ms stride).

| Train Mods. | PT | Mod. Drop | ASR WER Val | Test | AVSR WER Val | Test | VSR WER Val | Test |
|---|---|---|---|---|---|---|---|---|
| AV | - | N | 13.5 | 15.4 | **3.3** | **4.5** | **56.3** | **66.7** |
| AV | - | Y | **4.2** | **3.0** | 7.8 | 9.6 | 66.4 | 74.6 |
| A | - | - | **2.4** | **2.4** | 94.1 | 94.8 | 99.9 | 99.9 |
| V | - | - | 99.9 | 99.9 | **49.8** | **68.1** | **49.9** | **67.7** |
| AV | A | N | 3.2 | 3.2 | **1.9** | **2.5** | 67.1 | 71.3 |
| AV | A | Y | **2.9** | **2.6** | 2.8 | 2.6 | **60.7** | **67.0** |
| AV | V | N | 92.4 | 85.9 | **8.7** | **21.7** | **39.1** | **63.8** |
| AV | V | Y | **5.7** | **4.7** | 12.3 | 23.5 | 47.2 | 66.7 |

(d) AV-CPL (20ms stride) modality dropout w/ unlabeled VoxCeleb2 1,326h. The first row shows the seed model trained only on LRS3 with $p_m = p_a = 0.5$.

| Mod. Drop $p'_m = p'_a$ | VoxCeleb2 Unlabeled | ASR WER Val | Test | AVSR WER Val | Test | VSR WER Val | Test |
|---|---|---|---|---|---|---|---|
| Seed | Model | **2.9** | **2.6** | **2.8** | **2.6** | 60.7 | 67.0 |
| 0.5 | 1,326h | 3.4 | 2.2 | 3.1 | 2.0 | 46.8 | 55.7 |
| 0.25 | 1,326h | 4.0 | 2.7 | 3.9 | 2.5 | 34.0 | 49.9 |
| 0.1 | 1,326h | 4.9 | 3.3 | 4.9 | 3.0 | 30.9 | 48.4 |
| 0.05 | 1,326h | 5.8 | 4.1 | 5.5 | 4.0 | **28.2** | **48.3** |

model has more time slots to predict the correct tokens. Finally, AVSR performance is better with a stride of 20ms, and the final WER using characters and subwords is similar (9.6% vs 9.7%). Given these results, we use characters and a stride of 20ms to retain the best ASR and AVSR performance, which is useful for generating PLs on unlabeled videos.

**Modality Pre-Training.** We observed that it was difficult to train the audio-visual model jointly on both modalities from scratch. The VSR performance plateaued more rapidly than the AVSR and ASR performance. Moreover, the AVSR performance was usually worse than the ASR performance, and the ASR and VSR performance were usually worse than the single-modality baselines. To remedy this, we propose to pre-train the model on a single modality, and then start the training on the joint modalities with modality dropout. This can be viewed as a simple schedule on the modality dropout with $p_m = 0, p_a = 1$ at the beginning of training and arbitrary $p_m = p_a$ later. In Table 2c, we show the results for training jointly from scratch, followed by the results of training on only one modality. Next we show the results of training jointly when initialized from the model trained on one modality only. We find the best ASR (2.6%) and AVSR (2.6%) performance when the model is pre-trained on audio only, while the VSR (67.0%) performance nearly matches the best result. The AVSR performance of the model initialized from video-only pre-training is much worse (23.5%). Therefore, we first pre-train the model on audio-only data, and then train the model jointly on both modalities with modality dropout. With this pipeline, the AVSR performance matches the ASR performance, while the ASR and VSR performance are similar to the models trained separately on each modality (ASR: 2.6% vs 2.3% and VSR: 67.0% vs 65.0% for Transformer-Base.) We show the results of these experiments for the Base model on 30h, as well as the Large model on 433h/30h, in Appendix G. We note that such pre-training is less needed for the Large model on 433h, which shows that it is easier to learn from both modalities given enough data and parameters.

## 5.4 Audio-Visual Continuous Pseudo-Labeling

Once we train supervised audio-visual models on 433h or 30h of labeled LRS3 videos, we apply CPL and use the models to continuously generate PLs during training on unlabeled videos. While the modality dropout when training the seed model is $p_m = p_a = 0.5$, different modality dropout probabilities $p'_m = p'_a$ during CPL on unlabeled videos create a trade-off between ASR, AVSR, and VSR performance, as shown in Table 2d [1]. As the probability of using both modalities $p'_m$ decreases, the model is trained on more video-only data; VSR performance consistently improves up to the lowest $p'_m$ of 0.05 compared to the baseline trained without unlabeled videos. However, ASR and AVSR performance gets worse as $p'_m$ decreases, with the best performance at the initial modality dropout rate of 0.5. Given these observations, we present the main results using 433h of

---

[1] Training the seed model with different modality dropouts $p_m = p_a$ did not result in major differences.

Table 3: Comparison of our AV-CPL CTC encoder model (20ms stride, character tokens) with other works (most use S2S encoder-decoder transformers w/ or w/o CTC loss, 40ms stride and subword tokens) on 433h labeled LRS3 and 1,326h unlabeled VoxCeleb2 videos. "Separate FT" indicates models are fine-tuned *separately* for each task using 2-3× parameters.

| Method | Unlabeled Data | Labeled Data | Model | Encoder Size | Criterion | Separate FT | LM | VSR | ASR | AVSR |
|---|---|---|---|---|---|---|---|---|---|---|
| *Supervised* | | | | | | | | | | |
| Afouras et al. (2018a) | - | 1,519h | Transformer | - | S2S | - | ✗ | 58.9 | 8.3 | 7.2 |
| Xu et al. (2020a) | - | 590h | RNN | - | S2S | - | ✗ | 57.8 | 7.2 | - |
| Makino et al. (2019) | - | 31,000h | RNN | 63M | Transducer | ✓ | ✗ | 33.6 | 4.8 | 4.5 |
| Ma et al. (2021b) | - | 590h | Conformer | - | CTC+S2S | - | ✗ | 43.3 | 2.3 | 2.3 |
| Serdyuk et al. (2021) | - | 90,000h | Transformer | - | Transducer | - | ✗ | 25.9 | - | 2.3 |
| Serdyuk et al. (2022) | - | 90,000h | Conformer | 310M | Transducer | ✓ | ✗ | 17.0 | - | 1.6 |
| Chang et al. (2023) | - | 100,000h | Conformer | 570M | Transducer | ✓ | ✗ | 12.8 | - | 0.9 |
| ***Supervised** From Scratch (**Ours**)* | | | | | | | | | | |
| Base | - | 433h | Transformer | 96M | CTC | ✗ | ✓ | 67.0 | 2.6 | 2.6 |
| Large | - | 433h | Transformer | 315M | CTC | ✗ | ✓ | 58.6 | 2.7 | 3.1 |
| *Semi-Supervised Using **External** ASR Models* | | | | | | | | | | |
| Afouras et al. (2020) | 344h | 433h | CNN | - | CTC+S2S | - | ✗ | 59.8 | - | - |
| Ma et al. (2023) | 2,630h | 818h | Conformer | 186M | CTC+S2S | ✓ | ✗ | **19.1** | **1.0** | **0.9** |
| *Self-Supervised (Base Models)* | | | | | | | | | | |
| LiRA (Ma et al., 2021a) | 433h | 433h | Transformer | 103M | S2S | ✓ | ✗ | 49.6[1] | - | - |
| AV2Vec-lip (Zhang et al., 2023) | 433h | 433h | Transformer | 103M | S2S | ✓ | ✗ | 39.9 | - | 2.6 |
| AV-HuBERT (Shi et al., 2022a) | 433h | 433h | Transformer | 103M | S2S | ✓ | ✗ | 44.0 | - | 2.8[1] |
| RAVEn (Haliassos et al., 2023) | 433h | 433h | Transformer | 97M | CTC+S2S | ✓ | ✗ | 39.1 | 2.2 | - |
| AV-data2vec (Lian et al., 2023) | 433h | 433h | Transformer | 103M | S2S | ✓ | ✗ | **39.0** | **2.0** | **1.8** |
| AV-HuBERT (Shi et al., 2022a) | 1,759h | 433h | Transformer | 103M | S2S | ✓ | ✗ | 34.8 | - | 1.8[1] |
| RAVEn (Haliassos et al., 2023) | 1,759h | 433h | Transformer | 97M | CTC+S2S | ✓ | ✗ | 33.1 | 1.9 | - |
| VATLM (Zhu et al., 2023) | 1,759h[2] | 433h | Transformer | 107M | S2S | ✓ | ✗ | 34.2 | - | 1.7 |
| AV-data2vec (Lian et al., 2023) | 1,759h | 433h | Transformer | 103M | S2S | ✓ | ✗ | **32.9** | **1.7** | **1.4** |
| *Self-Supervised (Large Models)* | | | | | | | | | | |
| AV-HuBERT (Shi et al., 2022a) | 433h | 433h | Transformer | 325M | S2S | ✓ | ✗ | 41.6 | - | 2.5[1] |
| AV-data2vec (Lian et al., 2023) | 433h | 433h | Transformer | 325M | S2S | ✓ | ✗ | **37.4** | **1.9** | **1.7** |
| AV-HuBERT (Shi et al., 2022a;b) | 1,759h | 433h | Transformer | 325M | S2S | ✓ | ✗ | 28.6 | 1.6 | 1.4 |
| RAVEn (Haliassos et al., 2023) | 1,759h | 433h | Transformer | 671M | CTC+S2S | ✓ | ✗ | 27.8 | **1.4** | - |
| VATLM (Zhu et al., 2023) | 1,759h[2] | 433h | Transformer | 332M | S2S | ✓ | ✗ | 28.4 | - | **1.2** |
| u-HuBERT (Hsu & Shi, 2022) | 1,759h | 433h | Transformer | 325M | S2S | ✗ | ✗ | 29.1 | 1.5 | 1.3 |
| u-HuBERT* (Hsu & Shi, 2022) | 1,759h[2] | 433h | Transformer | 325M | S2S | ✗ | ✗ | **27.2** | **1.4** | **1.2** |
| AV-data2vec (Lian et al., 2023) | 1,759h | 433h | Transformer | 325M | S2S | ✓ | ✗ | 28.5 | **1.4** | 1.3 |
| *Self-Supervised + Self-Training with **External** ASR Models (Large Models)* | | | | | | | | | | |
| AV-HuBERT (Shi et al., 2022a) | 1,759h | 433h | Transformer | 325M | S2S | ✓ | ✗ | 26.9 | - | - |
| RAVEn (Haliassos et al., 2023) | 1,759h | 433h | Transformer | 671M | CTC+S2S | ✓ | ✓ | **23.1** | **1.4** | - |
| VATLM (Zhu et al., 2023) | 1,759h | 433h | Transformer | 332M | S2S | ✓ | ✗ | 26.2 | - | **1.2** |
| ***AV-CPL:** Continuous Pseudo-Labeling (**Ours**)* | | | | | | | | | | |
| Base (Mod. Drop $p'_m = p'_a = 0.1$) | 1,326h | 433h | Transformer | 96M | CTC | ✗ | ✓ | 48.4 | 3.3 | 3.0 |
| Base (Mod. Drop $p'_m = p'_a = 0.5$) | 1,326h | 433h | Transformer | 96M | CTC | ✗ | ✓ | 55.7 | **2.2** | **2.0** |
| Large (Mod. Drop $p'_m = p'_a = 0.1$) | 1,326h | 433h | Transformer | 315M | CTC | ✗ | ✓ | **45.3** | 3.0 | 3.4 |
| Large (Mod. Drop $p'_m = p'_a = 0.5$) | 1,326h | 433h | Transformer | 315M | CTC | ✗ | ✓ | 47.4 | 2.3 | 2.2 |

[1] LiRA Results reported by Shi et al. (2022a) and AV-HuBERT AVSR results reported by Lian et al. (2023).
[2] VATLM uses extra 3846h audio, 452h audio-text and 600M text data, and u-HuBERT* uses extra 452h audio (unlabeled) data.

labeled LRS3 videos in Table 3 with both 0.5 and 0.1 modality dropout, where 0.5 dropout obtains the best ASR and AVSR results, while 0.1 dropout obtains nearly the best VSR performance without a significant decrease in the ASR and AVSR performance. We present the results on 30h of labeled LRS3 data in Table 4 with 0.1 modality dropout to focus on improving the VSR performance.

In Table 3, we compare AV-CPL to other semi-supervised methods using 433h of labeled data. We also show supervised audio-visual methods trained with non-public data. With modality dropout of 0.1 during CPL, our method is able to significantly improve the VSR performance compared to the baseline trained only on the labeled data (58.6% → 45.3%) while maintaining near-optimal ASR (3.0%) and AVSR (3.4%) performance. Compared to the video-only CPL results in Table 1 (60.6% → 55.9%), the best VSR performance from AV-CPL (45.4%) is better by 10.5% absolute WER, which shows the advantage of using both audio and video inputs for pseudo-labeling, as opposed to using video-only inputs. With modality dropout of 0.5 during CPL, the improvement on VSR is not as large (58.6% → 47.4%), but the ASR and AVSR performance is improved over the baseline trained only on the labeled data (ASR: 2.6% → 2.3%, AVSR: 2.6% → 2.2%). We show the full results of our models on the validation set and with greedy decoding in the Appendix, Table H1.

Our best result for ASR (2.2%) and AVSR (2.0%) is close to the SSL state-of-the-art (1.4% and 1.2%), while our best result for VSR (45.3%) is worse than the state-of-the-art (27.2%). However, our method has several major advantages compared to the SSL methods. Our models are trained **jointly** on audio-visual data and **can perform AVSR, VSR, and ASR with a single trained model**, while the *SSL models are fine-tuned separately for each task* and require 3x more parameters to accomplish the same number of tasks (except for u-HuBERT (Hsu & Shi, 2022), which is also

Table 4: Comparison of our AV-CPL CTC encoder model ($p'_m = p'_a = 0.1$, 20ms stride and character tokens) with other works (most use S2S encoder-decoder transformers w/ or w/o CTC loss, 40ms stride and subword tokens) on 30h labeled LRS3 videos. "Separate FT" indicates models are fine-tuned *separately* for each task using 2-3× parameters.

| Method | Unlabeled Data | Labeled Data | Encoder Size | Criterion | Separate FT | LM | PL stage | LRS3 Test WER (%) VSR | ASR | AVSR |
|---|---|---|---|---|---|---|---|---|---|---|
| *Supervised From Scratch (**Ours**)* | | | | | | | | | | |
| Base | - | 30h | 96M | CTC | ✗ | ✓ | - | 94.3 | 8.9 | 11.4 |
| Large | - | 30h | 315M | CTC | ✗ | ✓ | - | 87.0 | 10.0 | 9.7 |
| *Self-Supervised (Base Models)* | | | | | | | | | | |
| Zhang et al. (2022) | 433h | 30h | 60M | CTC | ✓ | ✗ | - | 67.8 | 10.9 | 9.1 |
| LiRA (Ma et al., 2021a) | 433h | 30h | 103M | S2S | ✓ | ✗ | - | 71.9[2] | - | - |
| AV2Vec-lip (Zhang et al., 2023) | 433h | 30h | 103M | S2S | ✓ | ✗ | - | **45.1** | - | 5.8 |
| AV-HuBERT (Shi et al., 2022a) | 433h | 30h | 103M | S2S | ✓ | ✗ | - | 51.8 | - | 4.7[2] |
| RAVEn (Haliassos et al., 2023) | 433h | 30h | 97M | CTC+S2S | ✓ | ✗ | - | 47.0 | 4.7 | - |
| VATLM (Zhu et al., 2023) | 433h[1] | 30h | 107M | S2S | ✓ | ✗ | - | 48.0 | - | **3.6** |
| AV-data2vec (Lian et al., 2023) | 433h | 30h | 103M | S2S | ✓ | ✗ | - | 45.2 | **4.4** | 4.2 |
| AV-HuBERT (Shi et al., 2022a;b) | 1,759h | 30h | 103M | S2S | ✓ | ✗ | - | 46.1 | 4.4 | 4.1 |
| RAVEn (Haliassos et al., 2023) | 1,759h | 30h | 97M | CTC+S2S | ✓ | ✗ | - | 40.2 | 3.8 | - |
| VATLM (Zhu et al., 2023) | 1,759h[1] | 30h | 107M | S2S | ✓ | ✗ | - | 42.6 | - | 3.4 |
| AV-data2vec (Lian et al., 2023) | 1,759h | 30h | 103M | S2S | ✓ | ✗ | - | **37.8** | **3.7** | **3.3** |
| *Self-Supervised (Large Models)* | | | | | | | | | | |
| AV-HuBERT (Shi et al., 2022a) | 433h | 30h | 325M | S2S | ✓ | ✗ | - | 44.8 | - | 4.2[2] |
| AV-data2vec (Lian et al., 2023) | 433h | 30h | 325M | S2S | ✓ | ✗ | - | **40.5** | **3.7** | **3.4** |
| AV-HuBERT (Shi et al., 2022a;b) | 1,759h | 30h | 325M | S2S | ✓ | ✗ | - | 32.5 | 3.8 | 3.3 |
| RAVEn (Haliassos et al., 2023) | 1,759h | 30h | 671M | CTC+S2S | ✓ | ✗ | - | 32.5 | 2.7 | - |
| VATLM (Zhu et al., 2023) | 1,759h[1] | 30h | 332M | S2S | ✓ | ✗ | - | 31.6 | - | **2.7** |
| AV-data2vec (Lian et al., 2023) | 1,759h | 30h | 325M | S2S | ✓ | ✗ | - | **30.8** | 2.7 | 2.7 |
| *Self-Supervised + Self-Training with **External ASR** Models (Large Models)* | | | | | | | | | | |
| AV-HuBERT (Shi et al., 2022a) | 1,759h | 30h | 325M | S2S | ✓ | ✗ | - | 28.6 | - | - |
| RAVEn (Haliassos et al., 2023) | 1,759h | 30h | 671M | CTC+S2S | ✓ | ✓ | - | **23.8** | 1.9 | - |
| VATLM (Zhu et al., 2023) | 1,759h | 30h | 332M | S2S | ✓ | ✗ | - | 27.6 | - | **2.7** |
| ***AV-CPL**: Continuous Pseudo-Labeling (**Ours**)* | | | | | | | | | | |
| Base | 433h | 30h | 96M | CTC | ✗ | ✓ | PL LRS3 | 66.4 | 13.2 | 17.0 |
| Base | 1,759h | 30h | 96M | CTC | ✗ | ✓ | PL (Vox+LRS3) | 62.9 | **9.5** | **8.2** |
| Base | 1,326h | 30h | 96M | CTC | ✗ | ✓ | PL Vox | 63.3 | 14.4 | 13.6 |
| Base | 1,759h | 30h | 96M | CTC | ✗ | ✓ | +PL LRS3 | **57.3** | 10.8 | 13.6 |
| Large | 433h | 30h | 315M | CTC | ✗ | ✓ | PL LRS3 | 61.3 | 9.5 | 9.0 |
| Large | 1,759h | 30h | 315M | CTC | ✗ | ✓ | PL (Vox+LRS3) | 62.4 | **6.6** | **6.4** |
| Large | 1,326h | 30h | 315M | CTC | ✗ | ✓ | PL Vox | 63.1 | 15.4 | 12.2 |
| Large | 1,759h | 30h | 315M | CTC | ✗ | ✓ | +PL LRS3 | **56.7** | 10.0 | 10.4 |

[1] VATLM uses extra 3846h audio, 452h audio-text and 600M text data.  [2] Results reported by Shi et al. (2022a); Lian et al. (2023).

fine-tuned on audio-visual data and can perform all three tasks). Moreover, our models use only an encoder with beam-search decoding and a 4-gram LM while others use both an encoder and a decoder, which makes the total number of parameters of those models up to 1.5x the encoder size [2].

In Table 4, we compare AV-CPL to other methods using 30h of labeled data. Although directly performing AV-CPL on the combination of LRS3 and VoxCeleb2 works well and we obtain our best ASR (6.6%) and AVSR (6.4%) performance, we find that performing AV-CPL with VoxCeleb2 first followed by AV-CPL with LRS3 obtains better VSR performance, thus alleviating the domain mismatch between labeled and unlabeled data. AV-CPL significantly improves VSR performance compared to the baseline trained only on the labeled data (87.0% → 56.7%) and maintains practical ASR and AVSR performance. While training video-only models on just 30h of labeled videos resulted in >95% WER and thus video-only CPL was not possible, AV-CPL uses the seed model's strong AVSR performance (9.7%) to generate good PLs and confirms the advantage of multi-modal data. Moreover, our method performs all three tasks with *one model*, while all previous SSL methods presenting results on 30h labeled videos require a separate model for each task. We show the full results of our models on the validation set and with greedy decoding in the Appendix, Table H2.

## 6 CONCLUSION

We introduced audio-visual continuous pseudo-labeling for multi-modal semi-supervised learning. Our audio-visual models continuously generate pseudo-labels during training on unlabeled videos, which leads to significant improvements in VSR performance while maintaining practical ASR and AVSR performance. Our method uses a single objective for speech recognition throughout training and can perform ASR, VSR, AVSR with a single model. For future work, we would like to apply our method to more languages, especially since our method does not require external ASR models to generate pseudo-labels.

---

[2]For example, RAVEn's Transformer-Base decoder has half the parameters of the encoder.

ETHICS STATEMENT

The data used in this paper are publicly available for research purposes and were used under the following licenses: Creative Commons BY-NC-ND 4.0 license, Creative Commons Attribution 4.0 International License, and the TED terms of use. The datasets may have biases regarding racial, age, and gender attributes, which should be considered before deploying any models trained on them.

REPRODUCIBILITY STATEMENT

We provide implementation details and the full pseudo-code of our proposed method in the main paper and in the Appendix. We used datasets that are publicly available and include details such as dataset statistics and a discussion about performance on the validation set in Appendix. We report all of our results on the validation set for transparency and to make reproduction easier.

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

## A    Continuous Pseudo-Labeling Details

---

**Algorithm 1:** Audio-Visual Continuous Pseudo-Labeling (AV-SlimIPL)

---

**Data:** Labeled videos $L = \{(\boldsymbol{a}_i, \boldsymbol{v}_i), \boldsymbol{y}_i\}$ and unlabeled videos $U = \{(\boldsymbol{a}_j, \boldsymbol{v}_j)\}$
**Result:** Audio-visual model $\mathcal{M}_{\boldsymbol{\theta}}$
**Optional step:** Train $\mathcal{M}_{\boldsymbol{\theta}}$ on labeled audio-only data $L' = \{\boldsymbol{a}_i, \boldsymbol{y}_i\}$ and restart the optimizer;
1. Train $\mathcal{M}_{\boldsymbol{\theta}}$ on labeled audio-visual data $L = \{(\boldsymbol{a}_i, \boldsymbol{v}_i), \boldsymbol{y}_i\}$ with modality dropout $p_m = p_a$ until
   convergence, and restart the optimizer;                     ▷ Train the seed model.
2. Train $\mathcal{M}_{\boldsymbol{\theta}}$ on labeled audio-visual data $L = \{(\boldsymbol{a}_i, \boldsymbol{v}_i), \boldsymbol{y}_i\}$ with modality dropout $p'_m = p'_a$ for $M$
   steps ;              ▷ Begin CPL; "warm up" phase with new modality dropout.
3. **while** *cache is not full at size $C$* **do**
   | - Draw a random batch from $(\boldsymbol{a}, \boldsymbol{v}) \in U$;
   | - Generate its PL $\hat{\boldsymbol{y}}$ by $\mathcal{M}_{\boldsymbol{\theta}}(\boldsymbol{a}, \boldsymbol{v})$ with greedy decoding;
   | - Store $\{(\boldsymbol{a}, \boldsymbol{v}), \hat{\boldsymbol{y}}\}$ into the cache;
   | - Train $\mathcal{M}_{\boldsymbol{\theta}}$ on $L$ with augmentation and modality dropout $p'_m = p'_a$ for 1 update;
**end**
**repeat**
   | 4. Train $\mathcal{M}_{\boldsymbol{\theta}}$ on $L$ with augmentation and modality dropout $p'_m = p'_a$ for $N_L$ updates;
   | 5. **for** $N_U$ *updates* **do**
   |   | - Draw a random batch $B = \{(\boldsymbol{a}, \boldsymbol{v}), \hat{\boldsymbol{y}}\}$ from the cache;
   |   | - With probability $p$, $B$ is removed from the cache and replaced by a new unlabeled sample
   |   |   $(\boldsymbol{a}', \boldsymbol{v}') \in U$ and its PL $\hat{\boldsymbol{y}}'$ generated by current model state $\mathcal{M}_{\boldsymbol{\theta}}(\boldsymbol{a}, \boldsymbol{v})$;
   |   | - Train $\mathcal{M}_{\boldsymbol{\theta}}$ on batch $B$ with augmentation and modality dropout $p'_m = p'_a$ for 1 update;
   | **end**
**until** *convergence*;

---

---

**Algorithm 2:** Audio-Visual Continuous Pseudo-Labeling (AV-EMA-PL)

---

**Data:** Labeled videos $L = \{(\boldsymbol{a}_i, \boldsymbol{v}_i), \boldsymbol{y}_i\}$ and unlabeled videos $U = \{(\boldsymbol{a}_j, \boldsymbol{v}_j)\}$
**Result:** Audio-visual model $\mathcal{M}_{\boldsymbol{\theta}}$
**Optional step:** Train $\mathcal{M}_{\boldsymbol{\theta}}$ on labeled audio-only data $L' = \{\boldsymbol{a}_i, \boldsymbol{y}_i\}$ and restart the optimizer;
1. Train $\mathcal{M}_{\boldsymbol{\theta}}$ on labeled audio-visual data $L = \{(\boldsymbol{a}_i, \boldsymbol{v}_i), \boldsymbol{y}_i\}$ with modality dropout $p_m = p_a$ until
   convergence, and restart the optimizer;                     ▷ Train the seed model.
2. Initialize $\mathcal{M}_{\boldsymbol{\phi}} = \mathcal{M}_{\boldsymbol{\theta}}$ ;               ▷ Copy teacher weights from student.
**for** *$M$ steps;*          ▷ Begin CPL; "warm up" phase with new modality dropout.
**do**
   | - Train $\mathcal{M}_{\boldsymbol{\theta}}$ on labeled audio-visual data $L = \{(\boldsymbol{a}_i, \boldsymbol{v}_i), \boldsymbol{y}_i\}$ w/ modality dropout $p'_m = p'_a$ for 1 step;
   | - Update teacher weights: $\boldsymbol{\phi} \leftarrow \alpha\boldsymbol{\phi} + (1-\alpha)\boldsymbol{\theta}$
**end**
**repeat**
   | 4. Train $\mathcal{M}_{\boldsymbol{\theta}}$ on $L$ with augmentation and modality dropout $p'_m = p'_a$ for $N_L$ updates;
   | 5. **for** $N_U$ *updates* **do**
   |   | - Draw a random batch $(\boldsymbol{a}', \boldsymbol{v}') \in U$ and generate its PL $\hat{\boldsymbol{y}}'$ by $\mathcal{M}_{\boldsymbol{\phi}}(\boldsymbol{a}, \boldsymbol{v})$ with greedy decoding
   |   |   to form a batch $B = \{(\boldsymbol{a}, \boldsymbol{v}), \hat{\boldsymbol{y}}\}$;
   |   | - Train $\mathcal{M}_{\boldsymbol{\theta}}$ on batch $B$ with augmentation and modality dropout $p'_m = p'_a$ for 1 update;
   |   | - Update teacher weights: $\boldsymbol{\phi} \leftarrow \alpha\boldsymbol{\phi} + (1-\alpha)\boldsymbol{\theta}$
   | **end**
**until** *convergence*;

---

**AV-SlimIPL vs AV-EMA-PL.** The pseudo-code for AV-SlimIPL and AV-EMA-PL is shown in Algorithm 1 and Algorithm 2 respectively. AV-SlimIPL requires a data structure for maintaining the cache. The fastest method for doing this is to store the unlabeled samples and pseudo-labels in CPU memory. While this is practical for the original SlimIPL (Likhomanenko et al., 2021a) which only trains with audio, this is infeasible for video with cache sizes $C > 100$ due to the large size of the video frames. For example, a 1s spectrogram contains $80 \times 100 = 8,000$ values, while 1s of single-channel video contains $96 \times 96 \times 25 = 230,400$ values. Instead of keeping the samples in memory, a workaround is to maintain a mapping of unlabeled samples IDs in the dataset to their PLs. However, this requires loading the unlabeled data twice for each training step on un-

labeled data: once for pseudo-labeling and once for training on the unlabeled sample[3]. Loading video frames is significantly more time-consuming than loading audio due to the larger file sizes. In comparison, AV-EMA-PL only requires loading the unlabeled data once since the teacher model is used to generate PLs each time the student model is trained on a batch of unlabeled data. This requires keeping two copies of the model parameters, however, we find this to be easier on the memory and CPU thread consumption at the expense of being slightly slower than AV-SlimIPL due to the re-generation of PLs at every iteration. Therefore, we focused on AV-EMA-PL for our AV-CPL experiments.

## B   DATASET STATISTICS

Table B1: Dataset statistics for labeled LRS3 and unlabeled VoxCeleb2-English videos.

| Dataset | Split | # Samples | Audio duration (s) | | | | | | |
|---------|-------|-----------|------|------|------|-----|-----|-----|------|
| | | | Mean | Std. | Min. | 25% | 50% | 75% | Max |
| LRS3 | Test | 1,321 | 2.3 | 1.3 | 0.6 | 1.4 | 1.9 | 2.8 | 6.2 |
| LRS3 | Validation | 1,200 | 3.5 | 1.8 | 0.7 | 1.9 | 3.0 | 5.6 | 6.2 |
| LRS3, 30h | Train | 30,782 | 3.4 | 1.8 | 0.5 | 1.9 | 3.0 | 5.4 | 6.2 |
| LRS3, 433h | Train | 299,646 | 5.3 | 3.5 | 0.3 | 2.6 | 4.4 | 6.9 | 86.7 |
| VoxCeleb2 1,326h | Train | 628,418 | 7.6 | 3.8 | 0.4 | 4.9 | 6.3 | 9.0 | 22.5 |

Table B1 shows the number of samples and the length statistics for the sequences in the LRS3 and VoxCeleb2 dataset splits. The VoxCeleb2-English video split is provided by Shi et al. (2022a) according to an off-the-shelf English ASR model. We remove samples longer than 20s to ease the computational complexity.

## C   OPTIMIZATION DETAILS

We train all of the models up to 300k-400k steps on 8 A100 GPUs with 80GB of memory. Video samples with similar lengths are batched together such that the maximum number of frames is 5,680 frames (227s) per GPU. We apply SpecAugment (Park et al., 2019) during training to the input spectrograms with the following parameters: two frequency masks with frequency parameter $F = 30$, ten time masks with time mask parameter $T = 50$ and maximum time-mask ratio $p = 0.1$. When fine-tuning on the LRS3 30h training set, we use the same parameters except reduce the number of time masks to two since the videos are shorter on average. We use the AdaGrad optimizer (Duchi et al., 2011) with a learning rate of 0.03. The learning rate is warmed up for 64k steps and then held constant at 0.03 until 200k steps were reached. Then the learning rate is reduced by 2 every 50k updates if the WER does not improve on the validation set. Dropout and layer drop (Fan et al., 2020) are set to 0.1 during supervised training and CPL. Gradients are clipped to a maximum norm of 1.0. For AV-CPL and V-CPL experiments, we use $M = 5k$ warmup steps and $\alpha = 0.9999$. For the audio-only SlimIPL experiments, we use a cache size of 500 and $M = 20k$ warmup steps. Our implementation is in Jax (Bradbury et al., 2018).

## D   LANGUAGE MODEL INFERENCE

We train a 4-gram word-level language model on the LRS3 text using KenLM (Heafield, 2011). We use the Flashlight beam-search decoder (Kahn et al., 2022) implemented in Torchaudio (Yang et al., 2022) to integrate the language model. The perplexity on the LRS3 test set using the language model trained on the 433h training set was 92.5 excluding Out-of-Vocabularies (OOV) and 94.0 including OOV. The perplexity on the LRS3 test set using the language model trained on the 30h training set was 112.2 excluding OOV and 122.4 including OOV. We use the LRS3 text to construct a lexicon file which contains 51,292 words. We tuned the LM weight among $\{0, 1, 2, 4, 8\}$ and word insertion penalty among $\{\pm4, \pm2, \pm1, 0\}$ using grid search on the validation set and selected the LM weight of 2 and word insertion penalty of 0. We use a beam size of 1,500. We use the same LM decoding hyperparameters for all models.

---

[3]Smart data loading with proper pre-fetch is needed here.

# E    VALIDATION SET DISCUSSION

LRS3 (Afouras et al., 2018b) does not provide a validation set, therefore Shi et al. (2022a) randomly selected 1,200 samples (about 1h) from the 30h training set as the validation set. Several works since then have followed this setup (Haliassos et al., 2023; Zhu et al., 2023; Lian et al., 2023; Hsu et al., 2021a), however, so far no work has reported the performance of their final models on the validation set, except for an AV-HuBERT VSR ablation study (Shi et al., 2022a). We find it important to report the results on the validation set since the hyperparameters are tuned on the validation set with the test set held out until the final decoding. In most scenarios, the performance on the validation set is better than performance on the test set. However, for ASR, performance is better on the test set than on the validation set when using characters as the output units (Table G2). One interesting observation is that for VSR, the performance on the validation set is much better than the performance on the test set (Table F2), regardless of whether characters or subwords are used as the output units (Table G3). In some cases, the performance on the validation set is more than 20% absolute better than on the test set. Shi et al. (2022a) also report better VSR performance on the validation set compared to the test set by 9% absolute WER (Table D.1). Moreover, better performance on the validation set does not reliably indicate better performance on the test set. For example, the video-only V-CPL Base and Large models achieve 37.2% and 27.5% WER respectively on the validation set (Table F2) which is a significant difference, but they achieve 55.6% and 55.9% WER respectively on the test set, which is practically the same result. Upon further investigation, we found that the transcriptions for 1,044 of the 1,200 samples in the validation set are exact substrings of samples in the training set, while only 165 of the 1,321 samples in the test set are exact substrings of samples in the training set, which could potentially explain the discrepancy in performance on the sets and causes concern about over-fitting to particular sequences. Another reason could be that the test set may have more challenging visual conditions, for example, the test set may have faces shot at large angles, which would make VSR harder (Shillingford et al., 2019).

# F    AUDIO-ONLY AND VIDEO-ONLY CONTINUOUS PSEUDO-LABELING

Table F1: Comparison of audio-only semi-supervised methods: self-supervised learning and continuous pseudo-labeling. We reproduced SlimIPL (Likhomanenko et al., 2021a) on LRS3. The best results on the test set are bolded both with greedy decoding ("None") and with language model (LM) beam-search decoding (LM is trained either on 30h or 433h of LRS3 transcriptions). HuBERT (Hsu et al., 2021a) results presented by Shi et al. (2022a). All prior works use S2S encoder-decoder transformers w/ or w/o CTC loss except Ma et al. (2021b) used Conformer. RAVEn is from Haliassos et al. (2023).

| Method | Encoder Size | Criterion | Labeled Data | Unlabeled Data | PL Stage | LRS3 Val WER (%) | | | LRS3 Test WER (%) | | |
|---|---|---|---|---|---|---|---|---|---|---|---|
| | | | | | | None | LM 30h | LM 433h | None | LM 30h | LM 433h |
| *Supervised* | | | | | | | | | | | |
| RAVEn | 328M | CTC+S2S | 30h | - | - | - | - | - | **9.9** | - | - |
| SlimIPL (Ours) | 256M | CTC | 30h | - | - | 20.0 | 15.4 | 10.8 | 11.1 | 8.0 | **7.3** |
| E2E-Conformer | - | CTC+S2S | 433h | - | - | - | - | - | 2.3 | - | - |
| RAVEn | 328M | CTC+S2S | 433h | - | - | - | - | - | **2.2** | - | - |
| SlimIPL (Ours) | 256M | CTC | 433h | - | - | 3.4 | 3.1 | 2.1 | 2.5 | 2.2 | **2.1** |
| *Semi-Supervised* | | | | | | | | | | | |
| HuBERT | 300M | S2S | 30h | 433h | - | - | - | - | 4.5 | - | - |
| SlimIPL (Ours) | 256M | CTC | 30h | 433h | PL LRS3 | 11.1 | 8.8 | 6.1 | 5.2 | 3.6 | 3.2 |
| SlimIPL (Ours) | 256M | CTC | 30h | 433h | +FT LRS3 | 9.9 | 8.3 | 6.5 | **4.3** | 3.2 | **3.1** |
| HuBERT | 300M | S2S | 30h | 1,759h | - | - | - | - | **3.2** | - | - |
| SlimIPL (Ours) | 256M | CTC | 30h | 1,759h | PL (Vox+LRS3) | 12.2 | 9.0 | 5.9 | 5.7 | 3.9 | 3.3 |
| SlimIPL (Ours) | 256M | CTC | 30h | 1,326h | PL Vox | 12.2 | 9.4 | 6.2 | 6.3 | 4.3 | 3.9 |
| SlimIPL (Ours) | 256M | CTC | 30h | 1,759h | +PL LRS3 | 9.7 | 8.5 | 7.3 | 4.5 | 3.4 | 3.3 |
| SlimIPL (Ours) | 256M | CTC | 30h | 1,759h | +FT LRS3 | 9.4 | 8.4 | 7.4 | 3.8 | 3.2 | **3.0** |

In Table F1, we compare audio-based semi-supervised learning methods: HuBERT (Hsu et al., 2021a) as the SSL method and SlimIPL (Likhomanenko et al., 2021a) as the CPL method. We trained the SlimIPL method ourselves on LRS3 following the original model and hyperparameters. We report both the greedy and LM decoding results on both the LRS3 validation and test sets. We use the LRS3 30h training set as the labeled data, and either use the LRS3 433h training set as the unlabeled data or the combination of the LRS3 433h training data and VoxCeleb2 1,326h training data as unlabeled data. Comparing the supervised baselines, our model is able to match or outperform the reported state-of-the-art performance using a simple pipeline (encoder-only transformer with CTC loss compared to joint CTC and cross-entropy loss with a S2S encoder-decoder transformer).

Table F2: Comparison of video-only semi-supervised methods: self-supervised learning and continuous pseudo-labeling. AV-HuBERT (Shi et al., 2022a) results are for training with video-only and use S2S encoder-decoder transformers. The best results on the test set are bolded both with greedy decoding ("None") and with language model (LM) beam-search decoding (LM is trained either on 30h or 433h of LRS3 transcriptions).

| Method | Model | Encoder Size | Criterion | LRS3 Labeled | VoxCeleb2 Unlabeled | LRS3 Val WER (%) | | | LRS3 Test WER (%) | | |
|---|---|---|---|---|---|---|---|---|---|---|---|
| | | | | | | None | LM 30h | LM 433h | None | LM 30h | LM 433h |
| *Supervised* | | | | | | | | | | | |
| AV-HuBERT | Base | 103M | S2S | 30h | - | - | - | - | **94.3** | - | - |
| V-CPL | Base | 95M | CTC | 30h | - | 113.9 | 95.5 | 95.6 | 116.2 | 95.8 | **96.1** |
| AV-HuBERT | Base | 103M | S2S | 433h | - | - | - | - | **60.3** | - | - |
| V-CPL | Base | 95M | CTC | 433h | - | 64.1 | 55.2 | 50.6 | 73.6 | 65.1 | **65.0** |
| AV-HuBERT | Large | 325M | S2S | 30h | - | - | - | - | **92.3** | - | - |
| V-CPL | Large | 314M | CTC | 30h | - | 101.6 | 96.2 | 96.1 | 103.7 | 96.5 | **96.6** |
| AV-HuBERT | Large | 325M | S2S | 433h | - | - | - | - | **62.3** | - | - |
| V-CPL | Large | 314M | CTC | 433h | - | 41.3 | 35.7 | 32.4 | 66.0 | 61.1 | **60.6** |
| *Semi-Supervised with **External ASR** Models* | | | | | | | | | | | |
| AV-HuBERT | Large | 325M | S2S | 433h | 1,326h | - | - | - | **51.7** | - | - |
| *Semi-Supervised with Continuous Pseudo-Labeling (**Ours**)* | | | | | | | | | | | |
| V-CPL | Base | 95M | CTC | 433h | 1,326h | 51.3 | 43.3 | 37.2 | **63.7** | 56.4 | **55.6** |
| V-CPL | Large | 314M | CTC | 433h | 1,326h | 37.1 | 31.5 | 27.5 | **61.0** | 55.9 | **55.9** |

Comparing the semi-supervised methods, we find that SlimIPL can exceed HuBERT's performance. With 30 hours of labeled data and 433h of LRS3 unlabeled data, SlimIPL achieves 3.1% WER compared to HuBERT's 4.5% WER. Although directly performing CPL on the combination of LRS3 and VoxCeleb2 unlabeled data performs well, we find that performing CPL first on VoxCeleb2 and then on LRS3, followed by fine-tuning on the 30h labeled data in LRS3 works better and alleviates the domain mismatch between the labeled and unlabeled data. After these rounds of training on a total amount of 1,759h of unlabeled data from LRS3 and VoxCeleb2, SlimIPL achieves 3.0% WER compared to HuBERT's 3.2% WER. These results show that audio-only CPL methods transfer well to new datasets and are competitive with SSL methods, even with a simpler pipeline.

In Table F2, we show the full results of video-only continuous pseudo-labeling (V-CPL), including results with the Base model and results on the validation set. Our Base models achieve similar performance to the Base video-only AV-HuBERT trained from scratch without self-supervised learning, although our models use only an encoder with beam-search decoding and a 4-gram LM instead of a S2S encoder and transformer decoder. Applying V-CPL to the Base model, the WER with LM decoding is improved to 55.6%, which is even better than the Large model (55.9%). However, the Large model's greedy decoding performance (63.7%) is better than the Base model's (61.0%).

# G ABLATION STUDIES

Table G1: Ablation study on video-only continuous pseudo-labeling $\lambda$ (ratio of unsupervised to supervised updates). Experiments are conducted with LRS3 433h labeled video-only data and VoxCeleb2 1,326h unlabeled video-only data. We report greedy ("None") and beam-search decoding with a language model (LM) trained on 433h of LRS3 transcriptions.

| Transformer | $\lambda$ | LRS3 Val WER (%) | | LRS3 Test WER (%) | |
|---|---|---|---|---|---|
| | | No LM | LM 433h | No LM | LM 433h |
| Base | 1 / 1 | **51.3** | **37.2** | **63.7** | **55.6** |
| Base | 3 / 1 | 82.1 | 94.3 | 99.4 | 86.3 |
| Base | 1 / 3 | 68.6 | 56.1 | 77.2 | 67.9 |
| Large | 1 / 1 | 37.1 | 27.5 | 61.0 | **55.9** |
| Large | 3 / 1 | 40.8 | 29.7 | 65.9 | 59.3 |
| Large | 1 / 3 | **34.0** | **26.4** | **60.6** | 56.0 |

In Table G1, we study $\lambda = N_U/N_L$, the ratio of the number of unsupervised to supervised updates during video-only CPL (V-CPL). We find a ratio of 1 / 1 to work the best in most cases. We therefore adopt this ratio for the video-only and audio-visual CPL experiments.

In Table G2, we compare different combinations of output tokens and strides for the supervised ASR models (Likhomanenko et al., 2021a). We follow Shi et al. (2022a) to construct unigram-based subwords with a vocabulary size of 1k (Kudo, 2018). We use 433h of labeled audio from

Table G2: Ablation study on token set (characters and subwords) vs stride for audio-only ASR using 433h of labeled LRS3 audio and Transformer-Base. We report greedy ("None") and beam-search decoding with a language model (LM) trained on 433h of LRS3 transcriptions.

| Tokens | Stride | LRS3 Val WER (%) | | LRS3 Test WER (%) | |
|---|---|---|---|---|---|
| | | No LM | LM 433h | No LM | LM 433h |
| Characters | 20ms | **5.3** | **2.4** | **3.2** | **2.3** |
| Characters | 40ms | 12.5 | 7.0 | 7.4 | 5.3 |
| Subwords | 20ms | **3.7** | **3.3** | **8.5** | **6.9** |
| Subwords | 40ms | 4.6 | 3.7 | 8.9 | **6.9** |

LRS3 and the Transformer-Base model. The audio encoder is a convolutional layer with a kernel width of 7. Prior work keeps the video's native stride of 40ms and stacks 4 audio spectrogram frames to match the video frame stride (Shi et al., 2022a). However, in Table G2, we show that performance is always better with a 20ms stride using either characters or subwords as the output token. The best performance is obtained with character tokens and 20ms stride.

Table G3: Ablation study on token set (characters and subwords) for VSR. Experiments are conducted with LRS3 433h labeled video-only data and VoxCeleb2 1,326h unlabeled video-only data. We report greedy ("None") and beam-search decoding with a language model (LM) trained on 30h or 433h of LRS3 transcriptions.

| Tokens | LRS3 | VoxCeleb2 | LRS3 Val WER (%) | | | LRS3 Test WER (%) | | |
|---|---|---|---|---|---|---|---|---|
| | Labeled | Unlabeled | None | LM 30h | LM 433h | None | LM 30h | LM 433h |
| Characters | 30h | - | 113.9 | **95.5** | **95.6** | 116.2 | 95.8 | 96.1 |
| Subwords | 30h | - | **97.0** | **95.5** | 95.7 | **98.4** | **95.6** | **95.8** |
| Characters | 433h | - | 64.1 | **55.2** | 50.6 | 73.6 | **65.1** | 65.0 |
| Subwords | 433h | - | **57.2** | 55.6 | **45.3** | **70.7** | 65.4 | **64.9** |
| Characters | 433h | 1,326h | 51.3 | **43.3** | 37.2 | **63.7** | **56.4** | **55.6** |
| Subwords | 433h | 1,326h | **46.0** | 45.2 | **33.8** | 65.4 | 61.0 | 60.6 |

In Table G3 we compare characters to subwords as the output unit for the video-only model. We use the video's native stride of 40ms. Although subwords achieve better performance when training purely on labeled data, characters achieve significantly better performance when performing pseudo-labeling with unlabeled data (55.6% vs 60.6%).

We proposed to pre-train the audio encoder for supervised AVSR according to the results in Table 2c. We show the full results of such pre-training for the Transformer-Base model trained on 433h of labeled data, including results on the validation set and results with greedy decoding in Table G4. We show the results of these experiments for the Large model on 433h in Table G5, as well as the Base model on 30h in Table G6 and the Large model on 30h in Table G7. We note that such pre-training becomes less necessary for the Large model on 433h since the ASR, AVSR, and VSR performance is nearly the same both with and without pre-training, which shows that it is easier to learn from both modalities given enough data and representational power.

# H   AV-CPL FULL RESULTS

We show the full results of AV-CPL using 433h and 30h labeled LRS3 data including results on the validation set and with greedy decoding in Table H1 and Table H2.

Table G4: AVSR modality pre-training ablation with labeled LRS3 433h and Transformer-Base. We report greedy ("no LM") and beam-search decoding ("w/ LM") with a language model (LM) trained on 433h of LRS3 transcriptions.

| Train Mods. | PT | Mod. Drop | ASR WER (%) | | | | AVSR WER (%) | | | | VSR WER (%) | | | |
|---|---|---|---|---|---|---|---|---|---|---|---|---|---|---|
| | | | Val | | Test | | Val | | Test | | Val | | Test | |
| | | | No LM | w/ LM | No LM | w/ LM | No LM | w/ LM | No LM | w/ LM | No LM | w/ LM | No LM | w/ LM |
| AV | - | N | 25.9 | 13.5 | 24.5 | 15.4 | **5.9** | **3.3** | **6.6** | **4.5** | 68.7 | 56.3 | 74.0 | 66.7 |
| AV | - | Y | **8.7** | **4.2** | **4.7** | **3.0** | 14.3 | 7.8 | 13.3 | 9.6 | 75.7 | 66.4 | 81.0 | 74.6 |
| A | - | - | **5.1** | **2.4** | **3.8** | **2.4** | 95.6 | 94.1 | 95.7 | 94.8 | 99.9 | 99.9 | 99.8 | 99.9 |
| V | - | - | 99.9 | 99.9 | 99.9 | 99.9 | 67.3 | 49.8 | 77.6 | 68.1 | 67.1 | 49.9 | 77.6 | 67.7 |
| AV | A | N | 6.0 | 3.2 | 4.8 | 3.2 | **3.6** | **1.9** | 3.7 | **2.5** | 76.9 | 67.1 | 79.6 | 71.3 |
| AV | A | Y | 5.6 | 2.9 | 3.6 | 2.6 | 5.6 | 2.8 | **3.4** | 2.6 | 70.7 | 60.7 | 74.9 | 67.0 |
| AV | V | N | 93.9 | 92.4 | 89.7 | 85.9 | **14.4** | **8.7** | **27.1** | **21.7** | 50.5 | 39.1 | 69.8 | 63.8 |
| AV | V | Y | **11.2** | **5.7** | **7.2** | **4.7** | 19.5 | 12.3 | 29.3 | 23.5 | 56.5 | 47.2 | 71.6 | 66.7 |

Table G5: AVSR modality pre-training ablation with labeled LRS3 433h and Transformer-Large. We report greedy ("no LM") and beam-search decoding ("w/ LM") with a language model (LM) trained on 433h of LRS3 transcriptions.

| Train Mods. | PT | Mod. Drop | ASR WER (%) | | | | AVSR WER (%) | | | | VSR WER (%) | | | |
|---|---|---|---|---|---|---|---|---|---|---|---|---|---|---|
| | | | Val | | Test | | Val | | Test | | Val | | Test | |
| | | | No LM | w/ LM | No LM | w/ LM | No LM | w/ LM | No LM | w/ LM | No LM | w/ LM | No LM | w/ LM |
| AV | - | N | 22.4 | 13.1 | 31.6 | 26.8 | **4.7** | **2.4** | **4.1** | **3.1** | 65.2 | 53.1 | 66.8 | 59.3 |
| AV | - | Y | **6.2** | **3.2** | **4.2** | **2.7** | 6.0 | 3.2 | 4.8 | **3.1** | 56.4 | 45.9 | 65.0 | 58.6 |
| A | - | - | 3.8 | 2.1 | 2.7 | 2.0 | 97.8 | 97.8 | 98.0 | 98.1 | 99.9 | 99.9 | 99.9 | 99.9 |
| AV | A | N | 5.8 | **2.9** | 4.4 | 3.3 | **4.5** | **2.3** | **3.5** | **2.7** | 78.6 | 71.0 | 79.9 | 73.7 |
| AV | A | Y | **5.6** | 3.0 | **3.9** | 3.0 | 5.4 | 2.9 | 3.7 | 3.0 | 60.6 | 49.8 | 66.2 | 58.6 |

Table G6: AVSR modality pre-training ablation with labeled LRS3 30h and Transformer-Base. We report greedy ("no LM") and beam-search decoding ("w/ LM") with a language model (LM) trained on 30h and 433h of LRS3 transcriptions.

| Train Mods. | PT | Mod. Drop | ASR WER (%) | | | | | | AVSR WER (%) | | | | | | VSR WER (%) | | | | | |
|---|---|---|---|---|---|---|---|---|---|---|---|---|---|---|---|---|---|---|---|---|
| | | | Val | | | Test | | | Val | | | Test | | | Val | | | Test | | |
| | | | No LM | 30h | 433h | No LM | 30h | 433h | No LM | 30h | 433h | No LM | 30h | 433h | No LM | 30h | 433h | No LM | 30h | 433h |
| AV | - | N | 89.0 | 86.3 | 62.1 | 86.3 | 59.2 | 57.5 | 83.1 | 78.0 | 46.3 | 78.0 | **46.1** | **44.2** | **99.1** | 99.3 | 98.1 | **99.3** | 98.4 | 98.2 |
| AV | - | Y | 52.4 | 38.0 | 32.0 | 40.4 | 27.2 | 25.9 | 74.5 | 65.2 | 63.1 | 64.6 | 53.7 | 53.1 | 100.5 | 95.0 | 95.0 | 102.0 | 95.3 | 95.5 |
| A | - | - | 22.1 | 16.6 | 12.4 | 13.5 | 8.8 | 7.9 | 48.4 | 39.2 | 34.3 | 37.0 | 29.1 | 28.2 | 99.9 | 99.9 | 99.9 | 99.9 | 99.9 | 99.9 |
| AV | A | N | 31.9 | 25.1 | 20.4 | 21.2 | 15.5 | 14.7 | **24.5** | **19.3** | **15.1** | **15.9** | **11.6** | **11.0** | 94.8 | 90.0 | 90.3 | 95.5 | **89.8** | **90.0** |
| AV | A | Y | 23.3 | 17.3 | 13.1 | **13.9** | **9.5** | **8.9** | 29.1 | 22.2 | 17.6 | 18.0 | 12.3 | 11.4 | 99.7 | 93.3 | 93.4 | 101.4 | 94.1 | 94.3 |

Table G7: AVSR modality pre-training ablation with labeled LRS3 30h and Transformer-Large. We report greedy ("no LM") and beam-search decoding ("w/ LM") with a language model (LM) trained on 30h and 433h of LRS3 transcriptions.

| Train Mods. | PT | Mod. Drop | ASR WER (%) | | | | | | AVSR WER (%) | | | | | | VSR WER (%) | | | | | |
|---|---|---|---|---|---|---|---|---|---|---|---|---|---|---|---|---|---|---|---|---|
| | | | Val | | | Test | | | Val | | | Test | | | Val | | | Test | | |
| | | | No LM | 30h | 433h | No LM | 30h | 433h | No LM | 30h | 433h | No LM | 30h | 433h | No LM | 30h | 433h | No LM | 30h | 433h |
| AV | - | N | 86.8 | 68.7 | 62.5 | 84.4 | 61.5 | 60.2 | 81.6 | 56.8 | 45.9 | 75.4 | 45.9 | 43.7 | 98.6 | 97.3 | 97.2 | 98.7 | 97.1 | 97.1 |
| AV | - | Y | 38.7 | 30.9 | 26.4 | 27.0 | 20.7 | 19.5 | 67.1 | 60.4 | 58.5 | 50.2 | 43.6 | 43.0 | 101.1 | 97.5 | 97.3 | 102.5 | 98.0 | 98.0 |
| A | - | - | 21.8 | 17.4 | 13.7 | 13.4 | 10.0 | 9.5 | 98.0 | 97.5 | 97.2 | 96.2 | 95.2 | 95.0 | 99.9 | 99.9 | 99.9 | 99.9 | 99.9 | 99.9 |
| AV | A | N | 25.0 | 20.0 | 16.4 | 16.0 | 12.0 | 11.6 | **21.1** | **17.1** | **13.7** | **12.8** | **9.6** | **9.0** | 96.0 | 90.0 | 90.3 | 95.5 | 88.7 | 88.6 |
| AV | A | Y | 22.3 | 18.2 | 14.4 | **14.0** | **10.6** | **10.0** | 22.1 | 18.1 | 14.4 | 13.4 | 10.1 | 9.7 | **93.8** | **86.9** | **86.9** | 95.1 | 87.1 | 87.0 |

Table H1: AV-CPL main results on LRS3 433h labeled videos reported on LRS3 val and test sets. The seed models use modality dropout $p_m = p_a = 0.5$. We report greedy ("no LM") and beam-search decoding ("w/ LM") with a language model (LM) trained on 433h of LRS3 transcriptions.

| Model | Mod. Drop | VoxCeleb2 Unlabeled | ASR WER (%) | | | | AVSR WER (%) | | | | VSR WER (%) | | | |
|---|---|---|---|---|---|---|---|---|---|---|---|---|---|---|
| | $p'_m = p'_a$ | | No LM | w/ LM | No LM | w/ LM | No LM | w/ LM | No LM | w/ LM | No LM | w/ LM | No LM | w/ LM |
| Base | Seed | - | **5.6** | **2.9** | **3.6** | **2.6** | **5.6** | **2.8** | 3.4 | 2.6 | 70.7 | 60.7 | 74.9 | 67.0 |
| Base | 0.1 | 1,326h | 9.9 | 4.9 | 4.8 | 3.3 | 9.2 | 4.9 | 4.6 | 3.0 | **44.5** | **30.9** | **55.8** | **48.4** |
| Base | 0.5 | 1,326h | 7.0 | 3.4 | **3.5** | **2.2** | 6.3 | 3.1 | **3.2** | **2.0** | 61.6 | 46.8 | 64.9 | 55.7 |
| Large | Seed | - | 6.2 | 3.2 | 4.2 | 2.7 | 6.0 | 3.2 | 4.8 | 3.1 | 56.4 | 45.9 | 65.0 | 58.6 |
| Large | 0.1 | 1,326h | 8.8 | 4.6 | 4.9 | 3.4 | 8.0 | 4.1 | 4.4 | 3.0 | **33.3** | **24.1** | **51.0** | **45.3** |
| Large | 0.5 | 1,326h | **5.1** | **3.0** | **3.0** | **2.3** | **4.8** | **2.8** | **3.2** | **2.2** | 46.0 | 34.7 | 54.3 | 47.4 |

Table H2: AV-CPL main results on LRS3 30h labeled videos reported on LRS3 val and test sets. The seed models use modality dropout $p_m = p_a = 0.5$ while the AV-CPL models use modality dropout $p'_m = p'_a = 0.1$. We report greedy ("no LM") and beam-search decoding with a language model (LM) trained on 30h and 433h of LRS3 transcriptions.

| Model | Unlabeled Hours | PL Stage | ASR WER (%) | | | | | | AVSR WER (%) | | | | | | VSR WER (%) | | | | | |
|---|---|---|---|---|---|---|---|---|---|---|---|---|---|---|---|---|---|---|---|---|
| | | | Val | | | Test | | | Val | | | Test | | | Val | | | Test | | |
| | | | No LM | 30h | 433h | No LM | 30h | 433h | No LM | 30h | 433h | No LM | 30h | 433h | No LM | 30h | 433h | No LM | 30h | 433h |
| Base | - | Seed | 23.3 | 17.3 | 13.1 | 13.9 | 9.5 | 8.9 | 29.1 | 22.2 | 17.6 | 18.0 | 12.3 | 11.4 | 99.7 | 93.3 | 93.4 | 101.4 | 94.1 | 94.3 |
| Base | 433h | PL LRS3 | 29.3 | 27.9 | 18.4 | 16.9 | 14.8 | 13.2 | 32.9 | 28.5 | 22.9 | 22.2 | 18.0 | 17.0 | 71.2 | 61.4 | 58.9 | 74.5 | 66.6 | 66.4 |
| Base | 1,759h | PL (Vox + LRS3) | **22.0** | 17.9 | **10.6** | **14.2** | **10.7** | **9.5** | **20.8** | **17.0** | **11.1** | **12.8** | **9.0** | **8.2** | 71.2 | 62.5 | 59.9 | 70.5 | 63.2 | 62.9 |
| Base | 1,326h | PL Vox | 28.4 | 24.6 | 18.0 | 20.6 | 15.1 | 14.4 | 30.6 | 24.8 | 20.9 | 19.2 | 14.3 | 13.6 | 74.1 | 66.3 | 65.0 | 71.7 | 63.5 | 63.3 |
| Base | 1,759h | + PL LRS3 | 24.7 | 20.6 | 13.4 | 15.2 | 12.0 | 10.8 | 26.6 | 22.9 | 18.3 | 17.9 | 14.4 | 13.6 | **61.6** | **51.9** | **49.1** | **65.3** | **57.3** | **57.3** |
| Large | - | Seed | 22.3 | 18.2 | 14.4 | 14.0 | 10.6 | 10.0 | 22.1 | 18.1 | 14.4 | 13.4 | 10.1 | 9.7 | 93.8 | 86.9 | 86.9 | 95.1 | 87.1 | 87.0 |
| Large | 433h | PL LRS3 | 19.0 | 16.1 | 11.3 | 12.9 | 10.2 | 9.5 | 18.9 | 16.0 | 11.7 | 12.2 | 9.6 | 9.0 | 58.2 | 50.1 | 47.2 | 67.8 | 61.4 | 61.3 |
| Large | 1,759h | PL (Vox + LRS3) | **17.4** | **13.9** | **8.9** | **9.8** | **7.1** | **6.6** | **17.6** | **14.3** | **9.0** | **9.5** | **7.4** | **6.4** | 66.4 | 60.1 | 57.5 | 68.1 | 62.5 | 62.4 |
| Large | 1,326h | PL Vox | 33.1 | 19.7 | 20.8 | 19.7 | 16.2 | 15.4 | 28.6 | 16.6 | 17.1 | 16.6 | 12.8 | 12.2 | 74.1 | 70.2 | 64.3 | 70.2 | 63.2 | 63.1 |
| Large | 1,759h | +PL LRS3 | 19.6 | 17.2 | 11.5 | 13.1 | 10.8 | 10.0 | 20.3 | 20.3 | 12.7 | 13.3 | 13.1 | 10.4 | **54.9** | **48.1** | **43.3** | **63.1** | **57.5** | **56.7** |

