# OpenReview forum: "AV-CPL: Continuous Pseudo-Labeling for Audio-Visual Speech Recognition"
_ICLR.cc/2024/Conference — ICLR 2024 Conference Withdrawn Submission_

### Official Review · Reviewer_24HW · 2023-10-29

**Soundness:** 2 fair
**Presentation:** 2 fair
**Contribution:** 2 fair
**Rating:** 3
**Confidence:** 3

**Summary:**

This paper proposed AV-CPL for training a single model for ASR, VSP, and AVSP tasks by leveraging pseudo labels.

**Strengths:**

1. Detailed experiments and an ablation study.
2. Combining audio and video representations can outperform a single ASR in some scenarios.

**Weaknesses:**

The contribution of this paper is not very clear to me:
 1. Even though the authors claim that AV-CPL performs {AV, A, V}SR with a single model (compared to SSL models fine-tuned separately for each task), it actually exhibits significant variations in performance across these tasks due to different training strategies (e.g, modality dropout probability, PL stage)
 2. Considering that VSR is a less common and more challenging task than ASR, I have some reservations about the necessity of a 3-in-1 model that achieves approximately 40% WER on VSR, as opposed to having a dedicated VSR-specific model that can achieve around 20-30% WER for VSR alone.
 3. In Table 3, the difference in performance between AV-CLP and u-Hubert is not well explained, despite the similar training conditions (number of parameters, no separate FT).
4. The baseline, defined as solely trained on labeled data, is relatively weak.

**Questions:**

For table 3
 * It would be better to include the results of AV-CPL when leveraging 1757 hours of unlabeled data to align with u-Hubert's setup.
 * I suggest including the greedy-decoding results here, especially since most of the methods being compared in this context do not utilize a language model. This way, readers won't need to constantly switch between Table 3 and H1 to make performance comparisons.

---

### Official Review · Reviewer_iRHK · 2023-11-01

**Soundness:** 3 good
**Presentation:** 3 good
**Contribution:** 2 fair
**Rating:** 5
**Confidence:** 4

**Summary:**

This paper introduces continuous pseudo-labeling for audio-visual speech recognition (AV-CPL), a semi-supervised method to train an audio-visual speech recognition (AVSR) model on a combination of labeled and unlabeled videos with continuously regenerated pseudo-labels. It introduces two methods AV-SlimIPL and AV-EMA-PL for pseudo-labeling, respectively based on dynamic cache and a separate teacher model. The model is evaluated on LRS2 and LRS3 datasets and outperforms some baseline approaches in using unlabeled audio-visual speech data.

**Strengths:**

The method is capable of performing ASR, VSR, and AVSR using a single model, without the need for external ASR models. Additionally, the method is effective for using unlabeled audio-only and visual-only data. The paper is well-written, and the authors have conducted a thorough investigation of the training configuration, including architectural design, input stride, and output token set.

**Weaknesses:**

The model's performance lags behind several existing works in different settings, whether using unified or task-specific models. For instance, in the LRS3 433h regime (as shown in Table 3), the method significantly underperforms the state-of-the-art (VSR WER: 45.3 vs. 19.1, AVSR WER: 3.4 vs. 0.9). The model also demonstrates limited scalability, as can be seen from the marginal improvement from the Base to Large versions. Its advantage over SSL methods is also unclear.

**Questions:**

What is the performance of video-only models when incorporating more unlabeled video-only data in addition to VoxCeleb2?

---

### Official Review · Reviewer_2p3J · 2023-11-06

**Soundness:** 3 good
**Presentation:** 3 good
**Contribution:** 2 fair
**Rating:** 5
**Confidence:** 4

**Summary:**

This work tackles the problem of semi-supervised audio-visual speech recognition by introducing continuous pseudo-labeling on unlabeled samples. Experiments on LRS3 show its effectiveness.

**Strengths:**

The general structure is clear. The method is simple in general. It’s easy to follow.

**Weaknesses:**

The main focus of this work is to present the continuous pseudo-labeling strategy used in the learning process to introduce unlabeled data, but the specific manner to implement this center is similar to previous audio based CPL works (2021). One important point of how to prevent the model from degenerating is also similar to previous works, i.e. dynamic cache or EMA. The components involved are existing ones. Only be used for a new task, audio-visual speech recognition, instead of audio-only recognition?

A drawback of SSL methods, e.g. AV-HuBERT, claimed in this work is that they need to be finetuned separately for different tasks, i.e. 3xparameters for ASR, VSR, and AVSR separately. But we should also be able to introduce the simple random modality dropout operation in the finetune stage to perform finetune in a single stage. It’s not necessary to finetune 3 times for ASR, VSR, and AVSR separately.

**Questions:**

(1) I am a little confused about what’s new in this work, beyond using existing strategies for the audio-visual speech recognition task.
(2) In Table 1, the performance of introducing CPL with extra 1326h data is only slightly better than the original AV-HuBERT without extra data, i.e. 62.3 vs 61.1, and much worse than the semi-supervised version (about 10% worse than 51.7%). This small gap may illustrated the effectiveness but not much of the proposed methods in using unlabeled data.
(3) In table 3 and 4, the results show its improvement over the baseline, but show a too big gap from other works. I think this comparison could show the effectiveness, but not the superiority over others.

---

### Official Review · Reviewer_CBFK · 2023-11-06

**Soundness:** 3 good
**Presentation:** 3 good
**Contribution:** 2 fair
**Rating:** 3
**Confidence:** 4

**Summary:**

The paper presents the use of continuous pseudo-labeling (CPL) for the task of audio-visual speech recognition. Training with both labeled and unlabeled videos, pseudo labels are continuously generated to train a speech recognizer in a semi-supervised manner. The final recognizer can be used in AV, audio-only and video-only fashion. The results shown that a competitive AV-ASR system can be trained using CPL, without an external recognizer providing the labels, however the results do not match the best self-supervised results in the literature.

**Strengths:**

Clearly demonstrates how CPL can be used for AVSR with exhaustive experiments comparing against the literature with supervised, semi-supervised, self-training results. The method could be considered simpler or more hermetic in that only a single model is developed and used to generate the pseudo-labels compared to other semi-supervised results using external models with unknown provenance. They describe difficulties with using pre-trained AV models trained from scratch and suggest using a pre-trained audio-only model and then switch to modality drop out with CPL. This generates fairly good results when look at conducting AVSR, ASR and VSR evals with the same model.

**Weaknesses:**

Overall, the results seem incremental compared to the introduction of CPL for audio-only models with only experiments run on a speech recognition task. I'm of the opinion that this unfortunately greatly limits the strength of the contribution of the paper.

There seems to still be some issues with training large models with CPL. One would expect that the "Large" CPL trained model should work better than the "Base", but the final AVSR and ASR only best #s are for Base.

Finally, I would argue that AVSR task: 2.0% for CPL compared to 1.2% for state-of-the-art SSL are not close, and that self-supervised learning (without an external model) is a reasonable comparison for CPL (compared to 325M param u-Hubert). Even if there is an issue with the large model results, the ~100M param comparison with AV-data2vec is 30% relative off so it is hard to argue under what circumstances CPL would be preferable to SSL.

**Questions:**

Under what circumstances CPL would be preferable to SSL, vs AV-data2vec  or u-Hubert, given the results demonstrated in the paper?

What insights do you have on why the "Large" CPL models aren't doing significantly better than "Base" on the AVSR task?

Perhaps SSL is doing better than CPL because it is first using the unlabeled data to learn better low-level features, and then in the second fine-tuning step, it can build on it with good labels. Can CPL perhaps more incrementally learn lower level features first?

---

### Official Review · Reviewer_AEBL · 2023-11-09

**Soundness:** 3 good
**Presentation:** 3 good
**Contribution:** 2 fair
**Rating:** 3
**Confidence:** 3

**Summary:**

The paper experiments a continuous pseudo labeling (CPL) technique for semi-supervised audio-visual speech recognition (AVSR). Modality dropout is incorporated in the training which enables the system to perform inference in the presence of both audio and visual modalities, or only one modality. The main difference from existing audio-visual semi-supervised ASR methods is that the same model is used for supervised training and also for generating pseudo labels for the unsupervised data, instead of using a strong audio only model for generating the pseudo labels.

**Strengths:**

1. The literature survey is good.
2. Good ablation study for the choice of tokens, and modality fusion method are given.

**Weaknesses:**

1. The novelty is very weak. Like mentioned by the authors the work closely resembles Slim-IPL by Likhomanenko et al. and momentum pseudo labeling by Higuchi et al., with the addition of video modality being the only change. Applying CPL for AVSR is not novel either, the only narrow argument for novelty given by the authors is applying a different CPL method that was already established for audio only ASR. This is not a significant originality of idea nor execution.
2. In terms of performance, with either or both modalities it is not as good as many existing SOTA methods. The excuse given by the authors for this is that the proposed method could potentially use 33% lesser number of parameters because of lack of a decoder.

**Questions:**

1. In Section 3.1 it is said that "S2S models has issues during pseudo-label (PL) generation due to looping and over/under-generation" so CTC loss was used here. In Table 4, most methods using S2S outperforms your method, why is that. If S2S is an issue, why not use transducer loss?
2. It's not clear why in Table 2.B, character based ASR is so much better than subword based ASR given that most SOTA ASR systems these days use subwords as the token.

---

### Comment · Area_Chair_ekob · 2023-11-10
**reviewer-author discussions**

Dear All,

The reviewer-author discussion period will be from Nov. 10 to Nov. 22. For reviewers, please read the authors' responses and acknowledge it, respond to them early on in the discussion, and discuss points of disagreement. Thank you!

AC